# An Eco-Friendly Adsorbent Based on Bacterial Cellulose and Vermiculite Composite for Efficient Removal of Methylene Blue and Sulfanilamide

**DOI:** 10.3390/polym15102342

**Published:** 2023-05-17

**Authors:** Xiuzhi Bai, Zhongxiang Liu, Pengfei Liu, Yijun Zhang, Linfeng Hu, Tongchao Su

**Affiliations:** 1School of Chemistry and Chemical Engineering, Henan Institute of Science and Technology, Xinxiang 453003, China; amibai@126.com (X.B.); liuzhongxiang0903@126.com (Z.L.); zhangyijun@hist.edu.cn (Y.Z.);; 2School of Chemistry and Chemical Engineering, Hunan University of Science and Technology, Xiangtan 411201, China; 3Experiment and Test Center, Henan Institute of Science and Technology, Xinxiang 453003, China; 4School of Food Science, Henan Institute of Science and Technology, Xinxiang 453003, China; sutongchao66@163.com

**Keywords:** bacterial cellulose, vermiculite, biopolymers, water treatment, dye and antibiotic removal

## Abstract

In this work, a novel composite of bacterial cellulose (BC) and expanded vermiculite (EVMT) composite was used to adsorb dyes and antibiotics. The pure BC and BC/EVMT composite were characterized using SEM, FTIR, XRD, XPS and TGA. The BC/EVMT composite exhibited a microporous structure, providing abundant adsorption sites for target pollutants. The adsorption performance of the BC/EVMT composite was investigated for the removal of methylene blue (MB) and sulfanilamide (SA) from an aqueous solution. The adsorption capacity of BC/ENVMT for MB increased with increasing pH, while the adsorption capacity for SA decreased with increasing pH. The equilibrium data were analyzed using the Langmuir and Freundlich isotherms. As a result, the adsorption of MB and SA by the BC/EVMT composite was found to follow the Langmuir isotherm well, indicating a monolayer adsorption process on a homogeneous surface. The maximum adsorption capacity of the BC/EVMT composite was found to be 92.16 mg/g for MB and 71.53 mg/g for SA, respectively. The adsorption kinetics of both MB and SA on the BC/EVMT composite showed significant characteristics of a pseudo-second-order model. Considering the low cost and high efficiency of BC/EVMT, it is expected to be a promising adsorbent for the removal of dyes and antibiotics from wastewater. Thus, it can serve as a valuable tool in sewage treatment to improve water quality and reduce environmental pollution.

## 1. Introduction

With the rapid development of industrialization, there is an increasing need for the development of efficient technologies to reduce the environmental impact of wastewater to acceptable levels [1]. Dyes and antibiotics are both common pollutants produced by various industries [2]. Methylene blue (MB) is one such dye that is widely used in the printing, textile, paint, plastic and food industries, and is considered one of the main components of sewage [3]. Sulfonamides (SAs) are commonly used antibacterial agents in veterinary and human medicine, as well as in agriculture as herbicides [4]. High concentrations of MB and SA have been found in industrial wastewaters, surface and ground waters [5,6], both of which are toxic to living organisms. High exposure to MB may cause nausea, breathing difficulties, tissue necrosis, and liver and central nervous system problems. SA contains a strong basic group, which could induce eye, skin, and respiratory aggravation. Additionally, due to their origin and complex structure, dyes are not easily degraded chemically, biologically or by light. Similarly, sulfanilamide is also poorly biodegradable, leading to persistence in aquatic environments [7,8]. However, most conventional water treatment methods cannot efficiently remove MB or SA from wastewater [9]. Consequently, the development of effective techniques for removing dyes and antibiotics is crucial.

A variety of treatment techniques, including biodegradation, chemical oxidation, membrane filtration, coagulation, reverse osmosis and adsorption, are available for wastewater treatment [10]. Among these methods, adsorption is recognized as a promising technique due to its low cost, ease of design and operation, recyclability and high efficiency [11]. However, the use of hazardous precursors and the high operational costs associated with the synthesis of adsorbents remain obstacles to their practical applications. Therefore, research is underway to explore more economical and low-cost alternative adsorbents [12].

Currently, a variety of synthetic and natural adsorbents, such as activated carbon, zeolite, clay, cellulose, metal–organic frameworks (MOFs) and graphene have been used for adsorption and removal of pollutants, including dyes and antibiotics [13,14,15,16,17]. For instance, Prof. Imessaoudene’s group prepared activated carbon from date stones by chemical activation with phosphoric acid and used it for the removal of gallic acid. They also used zeolite, a low-cost, eco-friendly, and sustainable material to remove malachite green, MB and toxic Congo red dye [18,19,20,21]. Kyzas et al., utilized low-cost natural textile fibers as adsorbents to remove basic dye from synthetic effluents [22]. An MOF-modified bacterial cellulose/chitosan composite aerogel was also developed for the highly efficient removal of heavy metal ions and organic pollutants [23]. Cellulose can adsorb aqueous contaminants through physical and chemical interactions [24,25]. Due to numerous hydroxyl groups that provide active sites for functional modifications, cellulose has the potential to enhance the adsorption capability of adsorbents.

Bacterial cellulose (BC), being a polysaccharide, is widely acknowledged as an excellent absorbent due to its numerous outstanding properties [26,27]. BC exhibits unique physical and chemical properties, including an ultrafine network structure, a large surface area, high biocompatibility and excellent tensile strength, which qualifies this material for adsorption applications [28,29]. Therefore, BC-based adsorbents have been commonly applied in water purification technology through chemical procedures, cross-linking and physical composition [30,31,32]. In this study, an adsorbent was synthesized using two low-cost and environmentally beneficial materials, BC and expanded vermiculite (EVMT). EVMT is the product of vermiculite expansion, which has a loose and porous structure and soft texture. After heating, the volume increases and the density decreases [33]. EVMT is a mica-type silicate with exchangeable cations and active -OH groups on the surface. It is a widely used, cost-efficient adsorbent due to its layered crystalline structure, porous structure, hydrophilicity and high surface charge density [34,35]. According to some reported works, other materials such as MOF and graphene can be also used as adsorbents, but the production costs of these materials are very high [36,37]. By contrast, EVMT has an obvious cost advantage due to its cheap price. It has been demonstrated that the use of low-cost and environmentally friendly raw materials makes the new composites highly suitable for broad application prospects. Additionally, EVMT is chemically inert, durable, and environmentally safe, and has been widely used in various materials in recent years, such as nanocomposites, cement composites and coating [33,38,39]. Nevertheless, EVMT is a natural material that is difficult to directly use as a powder due to coagulation, flocculation and agglomeration [40]. Furthermore, it has been proven that the addition of a functional support material, such as eco-friendly BC, can significantly improve the adsorption properties of mineral clay [41]. Based on these considerations, the combination of EVMT with organic supporting polymers may take full advantage of BC and EVMT, which can produce a synergistic effect to significantly enhance the adsorption capacity of the BC/EVMT composite [42,43,44]. While previous studies have reported on BC and mineral clay composite materials, most of the mineral clay used was Ca–montmorillonite, and the composites were primarily used in the biomedical field [30].

The present research attempted to expand the environmental significance and application of an organic–inorganic composite as an adsorbent for water purification. A soft BC substrate was integrated with EVMT to prepare a BC/EVMT purifier. The resultant composite was used as a novel adsorbent for the removal of MB and SA, which are commonly found in industrial wastewater. The composite and its adsorption properties were characterized using FTIR, SEM, XRD, XPS and TGA. The adsorption efficiency of BC/EVMT for MB and SA in the presence of different pHs was investigated. Furthermore, equilibrium isotherms and adsorption kinetics were conducted to determine the reaction rate, adsorption capacity and possible removal mechanisms. This work provides a cost-efficient purifier for the removal of dyes and antibiotics.

## 2. Materials and Methods

### 2.1. Materials

Hipolypepton, yeast extract, glucose (C_6_H_12_O_6_) and mannitol were purchased from Beijing Aoboxing Bio-Tech Co., Ltd. (Beijing, China), magnesium sulfate heptahydrate and ethanol were purchased from Meryer Chemical Technology Co., Ltd. (Shanghai, China), and all were of analytical grade. Pure water was generated by a Milli-Q laboratory water purification system (Direct-Q 3, Millipore SAS, Molsheim, France). The microorganism used was *Acetobacterxylinum Gluconacetobacterxylinus*, which was purchased from Guangdong Microbial Culture Collection Center (Guangzhou, China). The standards methylene blue (99.7%) and sulfanilamide (99.9%) were purchased from Sigma-Aldrich (Guangzhou, China). Sodium hydroxide (97%), hydrochloric acid (37%) and ethanol (99.5%) were purchased from Shanghai Aladdin Reagent Co., Ltd. (Shanghai, China) and were all of analytic grade. The EVMT used in this study was purchased from Lingshou, Hebei province, China. The vermiculite (industrial grade) underwent calcination in a muffle furnace at 1000 °C for 10 min, then was precipitated and separated, naturally dried, machine ground, and sieved in 1000 mesh to obtain pure EVMT. The main phases of EVMT are EVMT and an EVMT–mica mixed-layer mineral [45]. It was composed of SiO_2_, Al_2_O_3_, MgO, K_2_O, Fe_2_O_3_, CaO, FeO, TiO_2_, and H_2_O, with Mg as the main interlayer cation.

### 2.2. BC Production and Purification

Bacterial culture was processed on seed culture medium, which was composed of 5 g/L hipolypepton, 5 g/L yeast extract, 5 g/L glucose, 5 g/L mannitol and 1 g/L magnesium sulfate heptahydrate, with a pH of 6.0. The medium was sterilized in an autoclave at 121 °C for 20 min. After cooling to room temperature, 5 mL/L of ethanol was added to the medium. Then, the *Acetobacterxylinum Gluconacetobacterxylinus* was transferred into the flask containing the liquid medium and statically cultivated at 30 °C for 15 days.

After cultivation, a layer of BC membrane was self-assembled onto the surface of the culture medium. The BC pellicle was treated with 0.1 mol/L sodium hydroxide solution at 80 °C for 3 h to remove lysed BC and medium components, followed by washing repeatedly with distilled water until the pH was neutral [30].

### 2.3. Preparation of BC/EVMT Composite

A 2% EVMT suspension (20 g EVMT mixed with 1 L deionized water) was prepared and stirred continuously at 25 °C for 2 h to obtain a homogenous mixture. Purified BC hydrogel was then sliced into squares (0.2 cm × 0.2 cm) and immersed in EVMT suspensions. The mixtures were stirred at 150 rpm and 30 °C for 24 h [46]. Further, the obtained BC/EVMT composite was extensively washed with distilled water to remove any excess residuals. Finally, sponge-like product samples were obtained after freeze-drying for 24 h, being maintained in the desiccator before further analysis.

### 2.4. Characterization Methods

The surface morphologies of all samples were studied by scanning electron microscopy (SEM). Prior to SEM observation, the samples were coated with spray-gold under high-vacuum conditions. The composition of BC and the BC/EVMT composite were analyzed using a PHI 5000 VersaProbe spectrometer (ULVAC-PHI, Chigasaki, Japan). FTIR spectra were reflected using a Thermo Scientific Nicolet iS5 spectrometer (Spectral range: 500 cm^−1^ to 4000 cm^−1^, resolution: 2 cm^−1^). The thermogravimetric analysis (TGA) was performed on a TA instrument (model no 300, EXSTAR, Fukuoka, Japan) with temperature ranging from 50 °C to 800 °C, a heating rate of 10 °C min^−1^, and under a nitrogen atmosphere. X-ray diffractometry (XRD) patterns were obtained using a Bruker AXS D8 advanced diffractometer with Cu Kα radiation (λ = 1.5418 Å) at 30 kV and 15 mA, with a scanning rate of 5° min^−1^ and a 2θ angle ranging from 5° to 70°. XPS analyses were carried out using X-ray photoelectron spectroscopy (ESCALAB 250Xi, Thermo Scientific Escalab, Waltham, MA, USA). The Brunauer–Emmett–Teller (BET) surface area and pore size were obtained by the N_2_ adsorption–desorption isotherm at 77 k with a Micromeritics ASAP 2020 volumetric adsorption analyzer.

### 2.5. Removal of Sulfanilamide and MB

In this study, SA and MB were selected as the target antibiotic and dye, respectively. The adsorption capacity of aerogels towards these compounds was investigated by a UV–visible spectrophotometer (U4100, Hitachi, Tokyo, Japan) at the wavelength of 666 nm for MB and 258 nm for SA, respectively. The batch adsorption experiments were operated in a 250 mL beaker. The typical procedure was conducted as below: BC/EVMT composite (20 mg, about 0.2 cm × 0.2 cm square) was added to 100 mL of MB or SA solution followed by agitation at 150 rpm and 30 °C for 120 min. The effect of pH on the interaction between MB/SA and the adsorbent was studied by adjusting the pH value (2, 4, 6, 8 and 10) using HCl (0.1 mol L^−1^) and/or NaOH (0.1 mol L^−1^). The removal efficiency of BC, EVMT, and BC/EVMT for MB and SA was also investigated, with initial adsorbate concentrations ranging from 10 to 80 mg L^−1^ in 100 mL of solution, and adsorbent doses ranging from 5 to 30 mg. For isotherm experiments, the initial concentrations of MB and SA ranged from 10 to 60 mg L^–1^, and the pH was adjusted to 8 and 2 for MB and SA, respectively. For kinetic studies, the initial concentrations of MB and SA were controlled at 60 mg L^–1^. The desired samples were collected at a given time from 0 to 120 min. The adsorption capacity at equilibrium (*q_e_*), the adsorption capacity at each time (*q_t_*), and the percentage removal were determined via the following equations [47,48]:(1)$$q_{e}=\frac{(C_{0}-C_{e})V}{m}$$
(2)$$q_{t}=\frac{(C_{0}-C_{t})V}{m}$$
(3)$$\%Removal=\frac{\left(C_{0}-C_{t}\right)}{m}\times 100$$
where *q_e_* (mg g^–1^) denotes the amount of MB or SA adsorbed onto the adsorbent at equilibrium, *q_t_* (mg g^–1^) shows the amount of adsorption at any time, *C*_0_ (mg L^–1^) and *C_e_* (mg L^–1^) are the initial and equilibrium adsorbate concentrations, respectively, and *C_t_* (mg L^–1^) displays the adsorbate concentration at any time (t). Additionally, *V* (L) denotes the volume of the adsorbate solution, and *m* (g) is the mass of dry adsorbent. The isotherms and kinetic studies were conducted using the same procedure as the equilibrium experiments. To ensure the reproducibility of the results, three trials were measured for all experiments, and the results were averaged.

## 3. Results and Discussion

### 3.1. Characterization

The morphological and microstructural characteristics of BC and BC/EVMT were analyzed using SEM, as depicted in Figure 1. It can be clearly seen from Figure 1a that pure BC displays a well-organized 3D network structure. Owing to the fine nano-reticular microfibrillar structure of BC, the expanded surface area and highly porous matrix of EVMT can promote efficient impregnation. In Figure 1b, it can be observed that in the BC/EVMT composite, EVMT particles are tightly attached to each fibril with slight agglomeration due to the coagulation, flocculation and agglomeration properties of powdered EVMT. The SEM characterization illustrated that the BC/EVMT composite was successfully synthesized. Additionally, the nano-porous architecture of the composite could provide an opportunity for the adsorption of various pollutants from aqueous solutions. Figure 1c,d illustrates that there was almost no change in the surface of BC/EVMT after MB or SA adsorption, indicating the excellent stability and potential applications of BC/EVMT in the removal of pollutants.

To further confirm the successful synthesis of the BC/EVMT composite aerogels, the FTIR spectra of BC, EVMT and the BC/EVMT composite aerogels were compared (Figure 2a). Notably, the strong characteristic peak appearing at 3350 cm^−1^ was attributed to the −OH group stretching vibration of the water molecules adsorbed on the surface of EVMT. The broad peak at 1640 cm^−1^ was attributed to the −OH group stretching vibration of water molecules adsorbed at the interlayer of EVMT [49,50]. These two peaks were present in both BC and BC/EVMT. The peak at 2920 cm^−1^ was caused by the aliphatic C−H vibration. The strong adsorptions at 1040 and 998 cm^−1^ were ascribed to the stretching of C−O−C (in BC) and Si−O−Si and Si−O−Al (in EVMT) [51,52], respectively. The above peaks can also be seen in the spectrum of the BC/EVMT composite, indicating the successful modification of EVMT on the surface of BC.

The XRD spectrum of EVMT particles and the BC/EVMT composite aerogels are given in Figure 2b. A strong peak appeared around 2θ = 22.79° in BC/EVMT, which was clearly attributed to BC pattern. The raw EVMT exhibited diffraction peaks at 2θ values of 27.33°, 34.49°, and 55.01° [40,53], which did not appear in BC. The same characteristic peaks were observed in the XRD of the BC/EVMT composite aerogels.

The surface elemental compositions of these samples were analyzed by XPS characterizations. The spectra in Figure 2c show that the intensity of the C1s peak weakened after grafting EVMT onto BC, and the content of C decreased from 60.46% to 53.97%. At the same time, the intensity of the O1s peak increased and the content of O increased from 34.06% to 37.11%. The Si2p peak appeared in BC/EVMT, but was not observed in bare BC. The Si2p peak can be split into three peaks at 101.73 eV (Si−OH groups), 102.75 eV (Si−O−M), and 103.69 eV (Si−O_4_ moiety), while the O1s peak can be split into three peaks at 530.31 eV (Si−O−Si), 531.74 eV (Si−OM), and 532.8 eV (SiO_4_ and Si−OH), confirming the presence of a silicate network in the adsorbent and indicating the existence of three types of linkages among the Si, O, and metal ions.

The thermal stabilities of BC and BC/EVMT were compared using TGA, and the results are presented in Figure 2d. During the analysis, BC exhibited an initial weight loss from 100 to 200 °C due to the loss of water. Then, a significant weight loss occurred at 200–400 °C, which may be attributed to the degradation of the main cellulose skeleton. This loss of biopolymer mass is caused by the depolymerization, dehydration, and decomposition of the glucose units in cellulose [54,55]. However, the thermal decomposition shifted significantly to a higher temperature (320−400 °C) for BC/EVMT. Notably, at the end of the weight loss process, BC had almost no weight left, while BC/EVMT had 57.36% weight left and no further mass loss occurred until 800 °C. These results demonstrate that BC/EVMT has better thermal stability than pure BC due to the addition of EVMT (a flame-retardant additive) [33]. It also could suggest that the residual undecomposed EVMT accounts for 57.36% of the weight of the BC/EVMT composite.

The BET specific surface areas of the samples were also studied to further compare their structures. The isotherms and BET data are shown in Appendix A. The isotherm of BC/EVMT exhibited type IV isotherms with an H1 loop (IUPAC classification) at P/P_0_ = 0.994, indicating that the prepared BC/EVMT had a mesoporous structure. The surface area of pure BC was 28.67 m^2^ g^−1^, while that of BC/EVMT increased to 55.70 m^2^ g^−1^. The pore size and pore volume of BC/EVMT were 17.48 nm and 0.3739 mL g^−1^, respectively. This may offer plenty of active sites and sufficient contact for MB and SA collection, which will be very helpful for enhancing the interaction with the adsorbates.

### 3.2. Adsorption Performance for Dyes and Antibiotics

#### 3.2.1. Effects of Physical and Chemical Parameters on Adsorption Effectiveness

The removal efficiency of MB and SA by BC, EVMT, and BC/EVMT was compared, as shown in Appendix A. Obviously, BC/EVMT had a greater adsorption capacity for both MB and SA than BC and EVMT. EVMT alone contributed less than 5% to the removal of both pollutants due to agglomeration. BC alone exhibited a slight removal of MB by 25.4%, and SA by 23.5%, which were much lower than those of the prepared BC/EVMT composite (72.7% for MB, 68.7% for SA).

The pH values can improve or decline the ionization of both the adsorbate and adsorbent, and can influence the surface charge of BC/EVMT. Additionally, the molecular structure of MB and SA may also change with the pH of the solution. Therefore, pH values play an important role in the adsorption process of MB and SA. The effect of pH on the removal of MB and SA was investigated at pH values ranging from 2 to 10 (Figure 3). The results indicated that the trends varied significantly between the adsorption of MB and SA. For MB adsorption, the adsorption capacity increased from around 35 to 75 mg/g as the pH increased from 2 to 10, which might be attributed to electrostatic interactions between the adsorbent and adsorbate. The presence of ions in aqueous solutions would also vary with pH changes. At lower pH, BC/EVMT was electro-positive due to the abundance of available protons, which hindered the adsorption of the positively charged cationic MB dye. Additionally, there was strong competition between MB^+^ and hydrogen ions (H^+^) for adsorbent sites. As the solution became more alkaline, the competition gradually decreased due to the increase in OH^-^ and decrease in H^+^. Furthermore, the number of negative charges on the adsorbent surface increased due to the increasing deprotonated silanol groups of EVMT, which enhanced the electrostatic attraction and adsorption capacity of the adsorbent for MB.

For SA adsorption, the uptake capacity decreased with increasing pH. SA is an amphoteric molecule with multiple ionizable functional groups, which can undergo protonation–deprotonation reactions under different solution pHs [12]. SA can exist as cationic, zwitterionic and anionic species according to the pH variation. At a pH of 2, almost all SA existed in the form of SA^+^, which had a strong electrostatic attraction with the adsorbents. As the pH increased from 2.0 to 6.0, SA mainly existed in the form of SA_0_, and there was a significant decrease in the adsorbed amount due to the gradual weakening of cation exchange. At neutral to alkaline conditions, further electrostatic repulsion occurred between SA^−^ and BV/EVMT. In summary, there was a significant change in the adsorption capacity for SA with the pH variation of the solution. This suggests that the electrostatic interaction is the dominant adsorption mechanism between SA and BV/EVMT.

The influence of other parameters such as the initial concentration of the adsorbate in the solution and the adsorbent dose were also studied. As shown in Appendix A, the removal efficiency for both MB and SA (from 94.5% to 52.0% for MB; from 82.6% to 45.7% for SA) decreased as the initial concentration increased (from 10 to 80 mg L^−1^). When the initial concentration is low, more active sites on the surface of the adsorbent are available, promoting higher adsorption. The effect of adsorbent dose is presented in Appendix A, where the removal efficiencies increased (from 28.1% to 87.9% for MB; from 25.6% to 83.2% for SA) as the amount of adsorbent increased (from 5 to 30 mg) due to the increased available adsorption sites.

#### 3.2.2. Adsorption Equilibrium Isotherms

Equilibrium studies provide valuable information on the adsorption capability of adsorbents. It also illustrates the interaction nature between adsorbed matter and the adsorbent. Generally, the Langmuir adsorption isotherm (Equation (4)) and the Freundlich adsorption isotherm (Equation (5)) are employed to analyze the equilibrium study of adsorption.
(4)$$\frac{C_{e}}{Q_{e}}=\frac{C_{e}}{Q_{m}}+\frac{1}{K_{L}Q_{m}}$$
(5)$$\ln Q_{e}=\ln K_{F}+n\ln C_{e}$$
where *C_e_* (mg/L) denotes the equilibrium concentration of MB or SA in an aqueous solution; *Q_e_* (mg/g) denotes the adsorption capacity of the adsorbent at equilibrium; *Q_m_* (mg/g) denotes the maximum adsorption capacity; *K_L_* (L/mg) is the Langmuir constant, representing adsorption heat in the adsorption process of the adsorbent; and *K_F_* (L/g) and *n* are the Freundlich parameters. In general, *n* > 1 represents that the adsorbent is favorable.

The fitting results of the isotherm models are illustrated in Figure 4a–d and a summary of the correlation parameters is provided in Table 1. Evidently, based on the correlation coefficient (*R^2^*), the Langmuir model provides better fits than the Freundlich model for MB and SA. This means that the adsorption of both pollutants on BC/EVMT is a homogeneous monolayer adsorption process [56]. Furthermore, the R_L_ values of MB and SA in Table 1 are less than 1, demonstrating that the adsorption of both pollutants on BC/EVMT is favorable. The prediction maximum adsorption capacity for the Langmuir equation was calculated to be 92.16 mg/g for MB and 71.53 mg/g for SA. The greater adsorption capacity for cationic MB than SA may be attributed to the negatively charged character of BC/EVMT.

#### 3.2.3. Adsorption Kinetics

The kinetic study can helpfully illustrate the physical or chemical interaction between the adsorbent and adsorbate. Additionally, it can provide important information for deducing the mechanism of the adsorption process and its efficiency [57]. The adsorption capacities of BC/EVMT for MB and SA as a function of contact time are shown in Figure 5. The experimental results indicated that the adsorption took place rapidly at the beginning stages of the adsorption process due to the large number of available sites on the surface of the adsorbent [41]. The adsorption equilibrium for MB and SA was achieved within 100 min and 20 min, respectively. It is worth noting that the adsorption rate order was significantly SA > MB. The slower adsorption of MB may be attributed to its higher molecular weight. Two kinetic models, the pseudo-first-order (Equation (6)) and pseudo-second-order (Equation (7)), were applied to fit the adsorption kinetics of the BC/EVMT aerogel for MB and SA using the following equations:(6)$$\ln(Q_{e}-Q_{t})=\ln Q_{e}-k_{1}t$$
(7)$$\frac{t}{Q_{t}}=\frac{1}{k_{2}Q_{e}{}^{2}}+\frac{t}{Q_{e}}$$
where *Q_e_* and *Q_t_* (mg/g) refer to the adsorption capacities at the equilibrium state and at time t (min), respectively. *k*_1_ (min^−1^) and *k*_2_ (g/(mg min)) denote the rate constant of the pseudo-first-order and pseudo-second-order kinetics, respectively. The adsorption kinetic curves of the MB dye and SA solution on BC/EVMT are shown in Figure 5a,b, respectively. The relevant parameters obtained from fitting the two kinetic models are presented in Table 2. 

As shown in Figure 5, both adsorption processes were better fitted to the pseudo-second-order model than the pseudo-first-order model due to larger correlation coefficients (*R^2^* > 0.99). The adsorption amounts at equilibrium (85.03 mg/g for MB and 75.07 mg/g for SA) estimated from the pseudo-second-order model were very close to the experimental values. Furthermore, the inset of Figure 5a,b clearly shows that the linear plots of t/Q_t_ versus t comply with pseudo-second-order kinetics, indicating that the chemical adsorption was involved in the MB and SA adsorption process. Some electron transfer may occur between the adsorbents and adsorbates [58].

### 3.3. Comparison of BC/EVMT with Other Related Adsorbents

Vermiculite- and BC-based adsorbents have previously been reported for the removal of pollutants such as dyes and antibiotics, and their adsorption capacities are summarized in Table 3. The adsorption capacity of MB and SA by different adsorbents was also compared. The as-prepared BC/EVMT composite demonstrated a competitive and satisfactory adsorption capacity and may have greater practical value in applications.

### 3.4. Application to Real Water Samples

To assess the feasibility and application of the prepared BC/EVMT composite in real water treatment, five water samples including tap water, river water, lake water, mineral water, and treatment plant effluent were analyzed by dissolving both MB and SA in the real water samples. Considering the acceptable difference in MB and SA removal, the tests of the real samples were carried out at a pH adjusted to 7.5, and the results are displayed in Table 4. Despite the differences in water quality, the BC/EVMT composite still exhibited effective removal behavior for all the real samples. For an aqueous solution containing 50 mg/L of MB and SA, the adsorption capacities of BC/EVMT for MB and SA were consistent with the removal performance of individual MB or SA adsorption in pure water. Furthermore, the adsorbent could be conveniently separated from the adsorption medium using tweezers for further use. Based on these results, it was confirmed that the BC/EVMT composite has significant potential for application in wastewater treatment.

## 4. Conclusions

In summary, this study aimed to prepare and characterize a composite adsorbent, BC/EVMT, for the efficient removal of cationic dyes and antibiotics from water. The prepared BC/EVMT was characterized using various techniques including SEM, FTIR, XRD, XPS, TGA and BET-specific surface area analysis using N_2_ adsorption. The specific surface area of BC/EVMT was found to be 55.70 m^2^/g, which is larger than that of pure BC and raw EVMT. The maximum adsorption capacity for MB and SA were found to be 92.16 mg/g and 71.53 mg/g, respectively, and the removal ratio was 72.7% for MB and 68.7% for SA (initial concentration, 60 mg/L; adsorbent dose, 20 mg). The adsorption process involved electrostatic attraction, cation exchange, and hydrogen bonding interactions. The experimental data were found to be better fitted to the Langmuir isotherm. The adsorption kinetics were well-described by the pseudo-second-order kinetics model. The thermodynamic studies suggested that the organic molecules were adsorbed onto the BC/EVMT composite by a monolayer coverage adsorption process, and chemo-adsorption plays the major role in the adsorption process. Overall, the developed BC/EVMT composite adsorbent shows promising potential as a candidate for removing cationic dyes and antibiotics from wastewater.

## Figures and Tables

**Figure 1 polymers-15-02342-f001:**
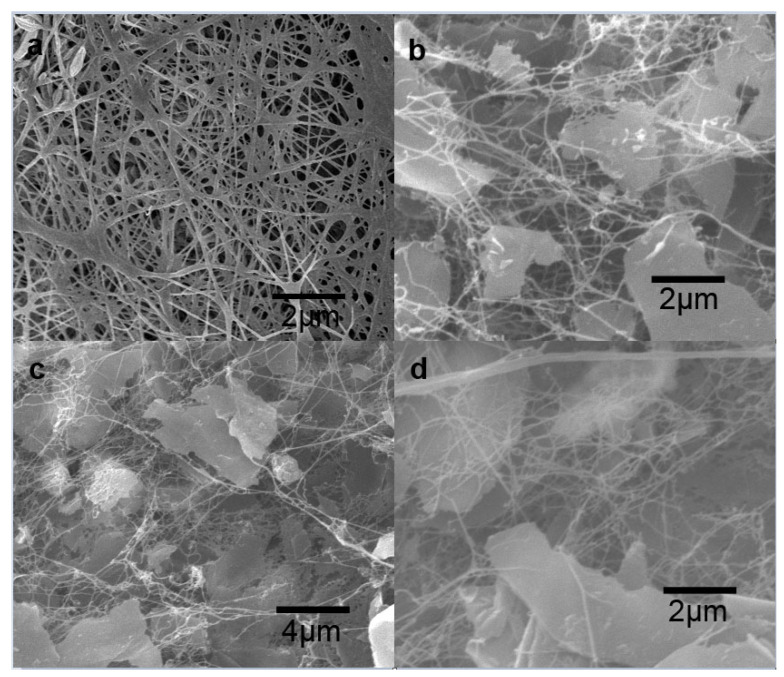
SEM images of (**a**) pure BC; (**b**) BC/EVMT; (**c**) BC/EVMT after MB adsorption; (**d**) BC/EVMT after SA adsorption (voltage: 5.0 kV for (**a**), 20.0 kV for (**b**–**d**); resolution: 5000× for (**a**), 6000× for (**b**–**d**)).

**Figure 2 polymers-15-02342-f002:**
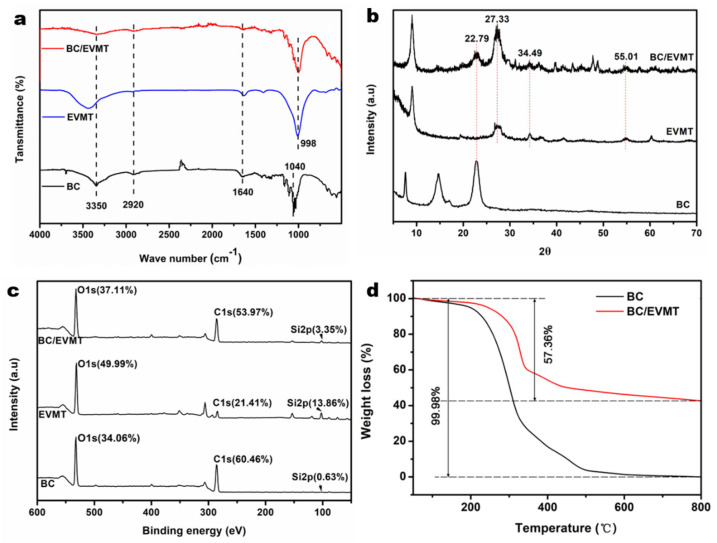
(**a**) FTIR spectra of BC, EVMT and BC/EVMT; (**b**) XRD spectra of BC, EVMT and BC/EVMT; (**c**) XPS spectra of BC, EVMT and BC/EVMT; (**d**) TGA curves of BC and BC/EVMT.

**Figure 3 polymers-15-02342-f003:**
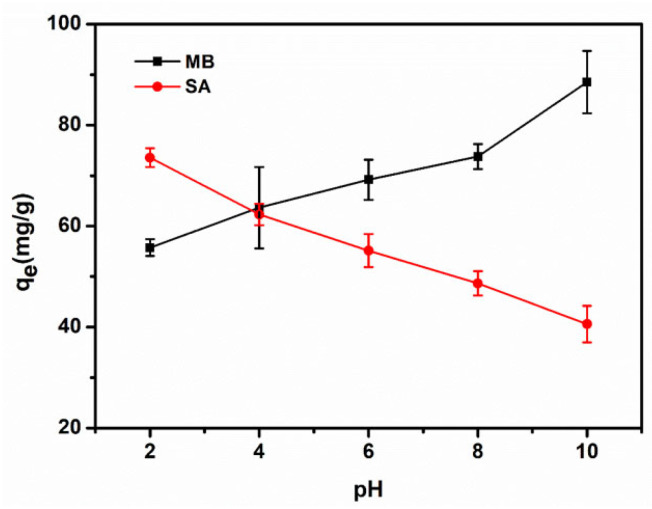
Effect of pH values on the adsorption of MB and SA by BC/EVMT (temperature: 30 °C, BC/EVMT adsorbent dosage: 20 mg, [MB]_0_: 50 mg/L, [SA]_0_: 50 mg/L).

**Figure 4 polymers-15-02342-f004:**
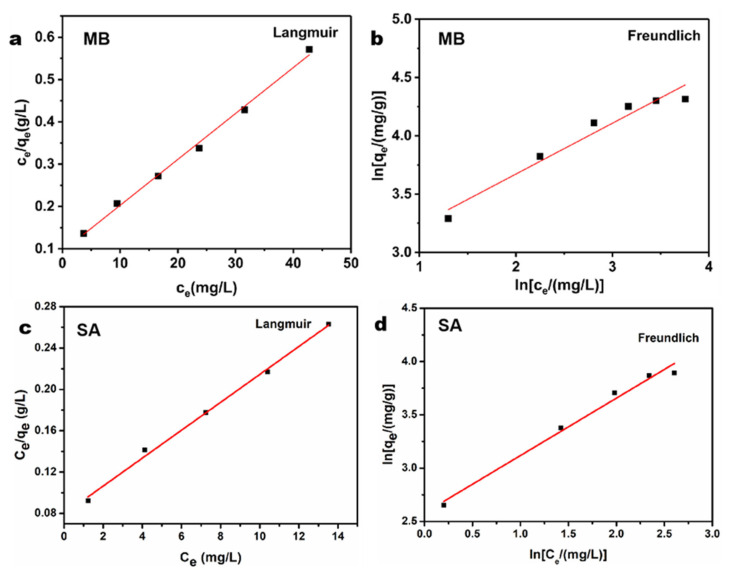
Adsorption isotherms of BC/EVMT towards (**a**,**b**) MB and (**c**,**d**) SA, and its harmony with Langmuir and Freundlich (temperature: 30 °C, pH_MB_: 8, pH_SA_: 2, adsorbent dosage: 20 mg, [MB]_0_: 10–60 mg/L, [SA]_0_: 10–60 mg/L).

**Figure 5 polymers-15-02342-f005:**
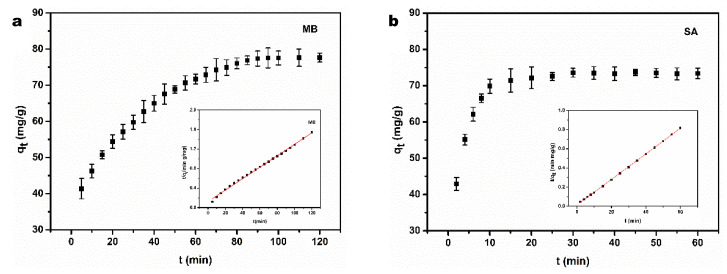
Effect of contact time and adsorption kinetics of BC/EVMT composite towards (**a**) MB and (**b**) SA, fitted by pseudo-second-order model (temperature: 30 °C, pH_MB_: 8, pH_SA_: 2, adsorbent dosage: 20 mg, [MB]_0_: 60 mg/L, [SA]_0_: 60 mg/L).

**Table 1 polymers-15-02342-t001:** Isotherm model parameters calculated from Langmuir and Freundlich for the removal of MB and SA by BC/EVMT.

	Langmuir			Freundlich		
	*Q_m_* (mg/g)	*K_L_* (L/mg)	*R* ^2^	*n*	*K_F_* (L/g)	*R* ^2^
MB	92.16	0.1151	0.9945	2.29	16.46	0.9455
SA	71.53	0.1862	0.9954	1.8437	13.17	0.9814

**Table 2 polymers-15-02342-t002:** Kinetic parameters for MB and SA removal on BC/EVMT composite.

	*C*_0_ (mg/L)	*q_e_*,_exp_ (mg/g)	Pseudo-First-Order Model			Pseudo-Second-Order Model		
			*k*_1_ (min^−1^)	*q_e_*,_cal_ (mg/g)	*R* ^2^	*k*_2_ × 10^3^ (g/ (mg min))	*q_e_*,_cal_ (mg/g)	*R* ^2^
MB	60	77.65	0.06081	100.7	0.8470	1.1244	85.03	0.9973
SA	60	73.37	0.1373	25.18	0.9738	12.8847	75.07	0.9997

**Table 3 polymers-15-02342-t003:** Comparison of maximum adsorption capacity and removal efficiency of previously reported related adsorbents.

Adsorbents	Adsorbates	Maximum Adsorption Capacity (mg/g)	Removal Efficiency (%)	References
Polyamide–Vermiculite	MB	76.42	99	[3]
Phosphatidylcholine–Vermiculite	oxytetracycline	66.40	77	[59]
	ciprofloxacin	93.72	96	
Gellan gum/bacterial cellulose	safranin	17.57	66	[60]
	crystal violet dye	13.49	28.5	
MOF-derived carbons	SA	256	not calculated	[61]
	chloroxylenol	588	not calculated	
BC/EVMT composite	MB	92.16	72.7	Present work
	SA	71.53	68.7	

**Table 4 polymers-15-02342-t004:** Adsorption amount (*Qe* (mg/g)) of MB and SA on BC/EVMT in the various real water samples (temperature: 30 °C, pH_MB_: 7.5, pH_SA_: 7.5, adsorbent dosage: 20 mg, [MB]_0_: 50 mg/L, [SA]_0_: 50 mg/L).

Real Samples	*Qe* (mg/g)	
	MB	SA
Tap water	70.25	48.33
River water	68.23	47.18
Lake water	67.45	45.77
Mineral water	65.11	47.21
Treatment plant effluent	60.39	41.91
Individual MB or SA in pure water	71.32	50.18

## Data Availability

The data that support the findings of this study are available from the corresponding authors upon reasonable request.

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
