# Peer review of "An Eco-Friendly Adsorbent Based on Bacterial Cellulose and Vermiculite Composite for Efficient Removal of Methylene Blue and Sulfanilamide"

_polymers, 2023, doi:10.3390/polym15102342_

Round 1

Reviewer 1 Report

Review of the manuscript ID: polymers-2326545; entitled: An eco-friendly adsorbent based on bacterial cellulose and vermiculite composites for efficient removal of dyes and antibiotics.   The work is focused on assessing the adsorption capacity of adsorbent materiel for the removal of dyes and antibiotics.

But there are major revisions to be made before likely publication as per the following comments
raised as below:

·         - The "Introduction" section needs a major rewrite. The following references are helpful in filling this gap: https://doi.org/10.3390/app12157587; https://doi.org/10.1080/01496395.2019.1676785, https://doi.org/10.1016/j.psep.2022.12.028, https://doi.org/10.3390/separations10010057

- The methodology used in this study should revised, because it has not been completely clarified.

- line 166,  « qe (mg g–1 ) is the amount of MB or SA adsorbed onto the adsorbent » qe is the amount of MB or Sa onto the adsorbent at equilibrium »

-It is better to present the Nadsorption isotherms at 77 K and pore size distribution (Bet). 

- The authors should add information about the number of replicates for all experiments (including preparation of the solid adsorbent material and MB concentration determination), hence the corresponding standard deviation. Then, all figures with experimental data points should contain their respective error bars.

- To determine the efficiency of the synthesized material, it is better to calculate the dye removal efficiency and compare it with data from the literature In order to elucidate the possible mechanism of dye removal, the equilibrium studies must be improved.

- The authors investigated only the linear regression adsorption models. By using the non-linear forms, the analysis of experimental data is more precise and accurate

- The adsorption isotherm model study discussion should be extended; there are more isotherms to investigate: Redlich-Peterson, Sips, Dubinin-Radushkevich, Temkin, etc in order to elucidate the adsorption mechanism

- Authors have to do XPS analysis in ordre to determine chemical compositions and states

- There are other significant parameters such as initial concentration, temperature, mass etc.

- Authors need to add sudy of Adsorption kinetics

-Authors have to add thermodynamic study

- The quality of the all figures need to be improved

- The conclusion section is very poor, the authors should rewrite this section according to the main objectives of the investigation.

Author Response

Reviewers' comments:

We appreciate the reviewer for his/her overall comments about the manuscript. We have addressed all the issues point-by-point in our response below.

Reviewer#1

Comments:

Review of the manuscript ID: polymers-2326545; entitled: An eco-friendly adsorbent based on bacterial cellulose and vermiculite composites for efficient removal of dyes and antibiotics.   The work is focused on assessing the adsorption capacity of adsorbent materiel for the removal of dyes and antibiotics.

But there are major revisions to be made before likely publication as per the following comments raised as below:

1)     - The "Introduction" section needs a major rewrite. The following references are helpful in filling this gap: https://doi.org/10.3390/app12157587; https://doi.org/10.1080/01496395.2019.1676785, https://doi.org/10.1016/j.psep.2022.12.028, https://doi.org/10.3390/separations10010057

Response: Thank the reviewer for your comments and advice. "Introduction" section was made a major rewrite. The four articles mentioned above are also cited. The added content could be seen in line 45-50, line 62 and 64-70 which was marked in red color.

The revised parts are as follows:

line 45-50:

High exposure to MB may cause nausea, breathing difficulties, tissue necrosis, liver and central nervous system. SA contains a strong basic group which could induce eye, skin, and respiratory aggravation. Additionally, due to the origin and complex structure, dyes are not easy to degrade chemically, biologically or by the light. Sulfanilamide is also poorly biodegradable and resulted it is persistent in aquatic environment.

line 62: zeolite,

line 64-70: For example, Prof. Imessaoudene’s group prepared an activated carbon from date stones by chemical activation with phosphoric acid and used for gallic acid removal. They also used zeolite as low-cost, ecofriendly and sustainable material to remove malachite green, MB and toxic Congo red dye [16–19]. Prof. Kyzas used very low-cost natural textile fibers as adsorbents to remove basic dye from synthetic effluents [20]. Metal-organic framework modified bacterial cellulose/chitosan composite aerogel was also developed for highly efficient removal of heavy metal ion and organic pollutant [21].

  1. Imessaoudene, A.; Cheikh, S.; Bollinger, J.C.; Belkhiri, L.; Tiri, A.; Bouzaza, A.; Jery, A. E.; Assadi, A.; Amrane, A.; Mouni, L. Zeolite waste characterization and use as low-cost, ecofriendly, and sustainable material for malachite green and methylene blue dyes removal: box–behnken design, kinetics, and thermodynamics. Appl. Sci. 2022, 12, 7587.
  2. Mammar, A. C.; Mouni, L.; Bollinger, J.C.; Belkhiri, L.; Bouzaza, A.; Assadi, A. A.; Belkacemi, H. Modeling and optimization of process parameters in elucidating the adsorption mechanism of Gallic acid on activated carbon prepared from date stones. Sep. Sci. Technol. 2019, 55 (17), 3113–3125.
  3. Bouchelkia, N.; Tahraoui H.; Amrane, A.; Belkacemi, H.; Bollinger, J.C.; Bouzaza, A.; Zoukel A.; Zhang, J.; Mouni, L. Jujube stones based highly efficient activated carbon for methylene blue adsorption: Kinetics and isotherms modeling, thermodynamics and mechanism study, optimization via response surface methodology and machine learning approaches. Process Saf. Environ. 2023, 170, 513–535.
  4. Imessaoudene, A.; Cheikh, S.; Hadadi, A.; Hamri, N.; Bollinger, J.C.; Amrane, A.; Tahraoui, H.; Manseri, A.; Mouni, L. Adsorption performance of zeolite for the removal of congo red dye: factorial design experiments, kinetic, and equilibrium studies. Separations. 2023, 10, 57.
  5. Kyzas, G.Z.; Christodoulou, E.; Bikiaris, D.N. Basic dye removal with sorption onto low-cost natural textile fibers. Processes. 2018, 6, 166.

2) - The methodology used in this study should revised, because it has not been completely clarified.

Response: Thank the reviewer for your comment. The methodology used in this study is supplemented in the revised manuscript.

The revised parts are as follows:

line 157-161: XPS analyses were performed on the X-ray photoelectron spectroscopy (ESCALAB 250Xi, Thermo Scientific Escalab, USA). The Brunauer–Emmett–Teller (BET) surface area and pore size were obtained by N2 adsorption-desorption isotherm at 77 k with a Micromeritics ASAP 2020 volumetric adsorption analyzer.

line 172-175: The removal efficiency of BC, EVMT, and BC/EVMT for MB and SA, both adsorbate initial concentrations (from 10 to 80 mg L-1) in solution (100 mL), and adsorbent dose (from 5 to 30 mg) were also studied.

line 184:   (3)

3) - line 166,  « qe (mg g–1 ) is the amount of MB or SA adsorbed onto the adsorbent » qe is the amount of MB or Sa onto the adsorbent at equilibrium »

Response: Thank the reviewer for your corrections. « qe (mg g–1) is the amount of MB or SA adsorbed onto the adsorbent » was replaced by « qe (mg g–1) is the amount of MB or SA onto the adsorbent at equilibrium » in the revised manuscript.

4) -It is better to present the N2 adsorption isotherms at 77 K and pore size distribution (Bet).

Response: Thank the reviewer for your comment. The N2 adsorption isotherms at 77 K and pore size distribution (Bet) were supplemented in the revised manuscript.

The revised parts are as follows:

Fig. S2 The nitrogen adsorption-desorption isotherms of EVMT, BC, and BC/EVMT by Brunauer-Emmett-Teller (BET) method.

Table S1 BET data of EVMT, BC, and BC/EVMT.

Samples

BET surface area

(m2/g)

BJH Adsorption

cumulative volume of pores (mL g-1)

Adsorption average

pore width (nm)

BC

28.67

0.1964

12.47

EVMT

11.50

0.0543

10.43

BC/EVMT

55.70

0.3739

17.48

Line 159-161:

“The Brunauer–Emmett–Teller (BET) surface area and pore size were obtained by N2 adsorption-desorption isotherm at 77k with a Micromeritics ASAP 2020 volumetric adsorption analyzer.”

Line 254-262:

The Brunauer–Emmett–Teller (BET) specific surface areas of the samples were also studied to further compare their structures. The isotherms and BET data was shown in Fig. S2 and Table S1. The isotherm of BC/EVMT exhibits type IV isotherms with an H1 loop (IUPAC classification) at P/P0=0.994, indicating that the prepared BC/EVMT is mesoporous structure. The surface area of pure BC is 28.67 (m2 g−1) while it increased to 55.70 (m2 g−1) of BC/EVMT. The pore size and pore volume of the BC/EVMT is 17.48 nm and 0.3739 mL/g, respectively. This may offer plenty of active sites and sufficient contact for MB and SA collection, which will be very helpful for enhancing interaction with adsorbates.

5) - The authors should add information about the number of replicates for all experiments (including preparation of the solid adsorbent material and MB concentration determination), hence the corresponding standard deviation. Then, all figures with experimental data points should contain their respective error bars.

Response: Thank the reviewer for reminding us. To ensure the reproducibility of the results, three individual experiments were measured for all experiments, and the results were averaged. The respective error bars of all experimental data points were added in the revised manuscript.

a

b

Fig. 5 Effect of contact time and adsorption kinetics of BC/EVMT towards a MB and b SA, fitted by pseudo-second-order model (temperature: 30 °C, pHMB: 8, pHSA: 2, adsorbent dosage: 20 mg, [MB]0: 60 mg/L, [SA]0: 60 mg/L).

Fig. S3 Removal efficiency of BC, EVMT, and BC/EVMT for MB and SA (temperature: 30 °C, pHMB: 8, pHSA: 2, adsorbent dosage: 20 mg, [MB]0: 60 mg/L, [SA]0: 60 mg/L).

Fig. S3 Effect of various experimental parameters on the adsorption of MB and SA by BC/EVMT: a MB and SA concentration and b adsorbent dosage. (temperature: 30 °C, pHMB: 8, pHSA: 2, adsorbent dosage: 20 mg, [MB]0: 60 mg/L, [SA]0: 60 mg/L, t: 120 min).

6) - To determine the efficiency of the synthesized material, it is better to calculate the dye removal efficiency and compare it with data from the literature. In order to elucidate the possible mechanism of dye removal, the equilibrium studies must be improved.

Response: Thank the reviewer for your comments and advice. The dye and antibiotic removal efficiency were calculated and compare it with data from the literature in the revised manuscript.

  1. The dye and antibiotic removal efficiency could be seen in line and Fig. S3 and the details described in line 265-271.

Fig. S3 Removal efficiency of BC, EVMT, and BC/EVMT for MB and SA (temperature: 30 °C, pHMB: 8, pHSA: 2, adsorbent dosage: 20 mg, [MB]0: 60 mg/L, [SA]0: 60 mg/L).

Line 265-271:

3.2.1 Effects of physical and chemical parameters on adsorption effectiveness

As depicted in Fig. S3, the removal efficiency of BC, EVMT, and BC/EVMT for MB and SA was compared. Obviously, the BC/EVMT displayed greater adsorption capacity for both MB and SA than BC and EVMT. The EVMT alone made almost no contribution to the removal of both pollutions (<5%) because of agglomeration. The BC alone exhibits a slight removal of MB by 25.4%, and SA by 23.5%, much lower than 72.7% for MB and 68.7% for SA, which removed by the prepared BC/EVMT.

  1. The dye and antibiotic removal efficiency compared with data from the literature could be seen in line 385-393 and Table 3.

Line 385-393:

3.3 Comparison of BC/EVMT with other related adsorbents

Vermiculite and BC based adsorbent were previously reported for removal of pollutant such as dye and antibiotic. The adsorption capacity of the similar materials was presented in Table 2. The adsorption capacity of MB and SA by different adsorbents was also compared. The as-prepared BC/EVMT displayed a competitive and satisfactory adsorption capacity and may have greater value in practical applications.

 Table 3 Comparison of maximum adsorption capacity and removal efficiency of previously reported related adsorbents.

Adsorbents

Adsorbates

Maximum Adsorption Capacity (mg/g)

Removal Efficiency (%)

References

Polyamide-Vermiculite

MB

76.42

99

3

Phosphatidylcholine-Vermiculite

oxytetracycline

66.40

77

49

ciprofloxacin

93.72

96

Gellan gum/bacterial cellulose

safranin

17.57

66

50

crystal violet dye

13.49

28.5

MOF-derived carbons

SA

256

not calculated

51

chloroxylenol

588

not calculated

BC/EVMT

MB

92.16

72.7

Present work

SA

71.53

68.7

  1. Liu, S.; Wu, P.; Yu, L.; Li, L.; Gong, B.; Zhu, N.; Dang, Z.; Yang, C. Preparation and characterization of organo-vermiculite based on phosphatidylcholine and adsorption of two typical antibiotics. Appl. Clay Sci. 2017, 137, 160–167.
  2. Nguyen, H. T.; Ngwabebhoh, F. A.; Saha, N.; Saha, T.; Saha, P. Gellan gum/bacterial cellulose hydrogel crosslinked with citric acid as an eco-friendly green adsorbent for safranin and crystal violet dye removal. Int. J. Biol. Macromol. 2022, 222, 77–89.
  3. Mondol, M. M. H. and Jhung S. H. Pore creation nanoarchitectonics from non-porous metal-organic framework to porous carbon for adsorptive elimination of sulfanilamide and chloroxylenol from aqueous solution. J. Hazard. Mater. 2022, 439, 129659.

7) - The authors investigated only the linear regression adsorption models. By using the non-linear forms, the analysis of experimental data is more precise and accurate. The adsorption isotherm model study discussion should be extended; there are more isotherms to investigate: Redlich-Peterson, Sips, Dubinin-Radushkevich, Temkin, etc in order to elucidate the adsorption mechanism.

Response: Thank the reviewer for your comment. The present discussion may support our statement, which can be further confirmed by several reported research works [1-3].

  1. Wang, W.; Zhao, W.; Zhang, H.; Xu, J.; Zong, L.; Kang, Y.; Wang, A. Mesoporous polymetallic silicate derived from naturally abundant mixed clay: A potential robust adsorbent for removal of cationic dye and antibiotic. Powder Technol. 2021, 390, 303-314.
  2. Arabkhani, P. and Asfaram, A. Development of a novel three-dimensional magnetic polymer aerogel as an efficient adsorbent for malachite green removal. Hazard. Mater. 2020, 384, 121394.
  3. Patil, C. S.; Gunjal, D. B.; Naik, V. M.; Jagadale, S. D.; Kadam, A. N.; Patil, P. S.; Kolekar, G. B.; Gore, A. H. Waste tea residue as a low cost adsorbent for removal of hydralazine hydrochloride pharmaceutical pollutant from aqueous media: An environmental remediation. J. Clean. Prod. 2019, 407-418.

8) - Authors have to do XPS analysis in ordre to determine chemical compositions and states.

Response: Thank the reviewer for your comment. The high-resolution XPS spectra of Si2p and O1s were supplemented in Fig. S1, and descried in the revised manuscript in line 229-234 as follows:

Line 231-236:

As shown in Fig. 4b and c, the Si2p peak can be split into three peaks at 101.73 eV (Si-OH groups), 102.75 eV (Si-O-M), and 103.69 eV (Si–O4 moiety), and the O1s peak can be split into three peaks at 530.31 eV (Si-O-Si), 531.74 eV (Si-OM), and 532.8 eV (SiO4 & Si-OH),which confirms the presence of a silicate network in the adsorbent, and there are three types of linkage among Si, O, and metal ions.  

Fig. S1 The high-resolution XPS spectra of Si2p (a) and O1s (b)

9) - There are other significant parameters such as initial concentration, temperature, mass etc.

Response: Thank the reviewer for your comment. The influence of other parameters such as initial concentration of adsorbate in the solution, and adsorbent dose were also studied in the revised manuscript. It can be found that the effect of temperature on the adsorption capacity was smaller in the temperature over 25 °C, so the effect of temperature on the adsorption capacity was not studied [1, 2].

Line 306-314:

The influence of other parameters such as initial concentration of adsorbate in the solution, and adsorbent dose were also studied. As shown in Fig. S4a, the removal efficiency (from 94.5% to 52.0% for MB, from 82.6% to 45.7% for SA) decreased with increasing the initial concentration (from 10 to 80 mg L-1) of MB and SA. When the initial concentration is low, more active sites on the surface of adsorbent are available, which promotes higher adsorption. The effect of adsorbent dose is presented in Fig. S4b. The removal efficiencies rise (from 28.1% to 87.9% for MB, from 25.6% to 83.2% for SA) with the number of adsorbent increases (from 5 to 30 mg), because of the available adsorption sites increased.

Fig. S4 Effect of various experimental parameters on the adsorption of MB and SA by BC/EVMT: (a) MB and SA concentration and (b) adsorbent dosage. (temperature: 30 °C, pHMB: 8, pHSA: 2, adsorbent dosage: 20 mg, [MB]0: 60 mg/L, [SA]0: 60 mg/L, t: 120 min).

  1. Arabkhani, P. and Asfaram, A. Development of a novel three-dimensional magnetic polymer aerogel as an efficient adsorbent for malachite green removal. Hazard. Mater. 2020, 384, 121394.
  2. Lin, K.-Y.A.; Lee, W.-D. Highly efficient removal of Malachite green from water by a magnetic reduced graphene oxide/zeolitic imidazolate framework self-assembled nanocomposite. Appl. Surf. Sci. 2016, 361, 114–121.

10) - Authors need to add study of Adsorption kinetics.

Response: Thank the reviewer for your comment. The study of Adsorption kinetics was expounded in original manuscript “3.2.3 Adsorption kinetics”

11) -Authors have to/. add thermodynamic study.

Response: Thank the reviewer for your comment. We have made every effort to follow the comments of the reviewers. Since time is limited, the thermodynamic study has not been completed. Additionally, thermodynamic experiments are classical but may not be necessary which can be further confirmed by several reported research works [1-4].

  1. Sirviö, J. A. and Visanko M. Lignin-rich sulfated wood nanofibers as high-performing adsorbents for the removal of lead and copper from water. Journal of Hazardous Materials. 2020, 383, 121174.
  2. Li, D.; Tian, X.; Wang, Z. Multifunctional adsorbent based on metal-organic framework modified bacterial cellulose/chitosan composite aerogel for high efficient removal of heavy metal ion and organic pollutant. Chem. Eng. J. 2020, 383, 123127.
  3. Li, Z.; Tian, H.; Yuan, Y.; Yin, X.; Wei, X.; Tang, L.; Wei, S. Metal-ion-imprinted thermo-responsive materials obtained from bacterial cellulose: synthesis, characterization, and adsorption evaluation. J. Mater. Chem. A 2019, 7, 11742–11755.
  4. Patil, C. S.; Gunjal, D. B.; Naik, V. M.; Jagadale, S. D.; Kadam, A. N.; Patil, P. S.; Kolekar, G. B.; Gore, A. H. Waste tea residue as a low cost adsorbent for removal of hydralazine hydrochloride pharmaceutical pollutant from aqueous media: An environmental remediation. J. Clean. Prod. 2019, 407-418.

12) - The quality of the all figures need to be improved.

Response: Thank the reviewer for your comment. The quality of the all figures were improved in the revised manuscript.

13) -The conclusion section is very poor, the authors should rewrite this section according to the main objectives of the investigation.

Response: Thank the reviewer for your comment. The conclusion section is added in line 413-416 and 417-418 of the revised manuscript.

Line 413-416:

The prepared BC/EVMT was explored via SEM, FTIR, XRD, XPS, TGA and BET-specific surface areas (N2 adsorption). The specific surface area of BC/EVMT is 55.70 m2/g, which is larger than that of pure BC and raw EVMT. 

Line 417-418:

The removal ratio is 72.7% for MB, and 68.7% for SA (initial concentration, 60 mg/L; adsorbent dose, 20 mg).

Reviewer 2 Report

The authors have presented an interesting and well written manuscript on the preparation of several composites based on bacterial cellulose and expand vermiculite, as adsorbent for dyes and antibiotics. The manuscript presented the synthesis and characterization of all products prepared, plus, the employed characterization techniques seem to be the most adequate, and results were well presented. In general, authors have well described the novelty of these new materials, however some accuracies must be rectified:

Line 138: Please describe the number of scans used in FT-IR analysis.

Line 141 and 142: Where is described that TGA analysis was conducted both under air (line 141) and nitrogen atmosphere (line 142)? Each one was performed in this study?

Line 162: Reference [32] do not refer the equations cited in the manuscript, could the authors provide a new reference that describes these equations?

Fig. 1 Please add SEM conditions in terms of resolution, voltage, etc.

Line 202: The authors should also describe raw BC characteristic peaks in were. Clearly, the peak at 22.79° from BC/EVMT is attributed to BC pattern.

Pag. 9: The authors should include in this section some adsorption capacity values, i.e., values reported in the literature from similar materials, in order to demonstrate the novelty of these new composites.

Author Response

Reviewer#2

Comments:

The authors have presented an interesting and well written manuscript on the preparation of several composites based on bacterial cellulose and expand vermiculite, as adsorbent for dyes and antibiotics. The manuscript presented the synthesis and characterization of all products prepared, plus, the employed characterization techniques seem to be the most adequate, and results were well presented. In general, authors have well described the novelty of these new materials, however some accuracies must be rectified:

1) Line 138: Please describe the number of scans used in FT-IR analysis.

Response: Thank the reviewer for your comment. The number of scans used in FT-IR analysis was described in the original manuscript in line 139 as follows: “FT-IR spectra were reflected by a Thermo Scientific Nicolet iS5 spectrometer (Spectral range: 500 cm1 to 4000 cm1, resolution: 2 cm1).”

2) Line 141 and 142: Where is described that TGA analysis was conducted both under air (line 141) and nitrogen atmosphere (line 142)? Each one was performed in this study?

Response: Thank the reviewer for reminding us. Our TGA analysis was conducted under nitrogen atmosphere. “in air atmosphere” has been deleted in the revised manuscript.

3) Line 162: Reference [32] do not refer the equations cited in the manuscript, could the authors provide a new reference that describes these equations?

Response: Thank the reviewer for reminding us. Reference [32] was updated with a new reference that describes these equations in the revised manuscript.

[37] Wang, W.; Zhao, W.; Zhang, H.; Xu, J.; Zong, L.; Kang, Y.; Wang, A. Mesoporous polymetallic silicate derived from naturally abundant mixed clay: A potential robust adsorbent for removal of cationic dye and antibiotic. Powder Technol. 2021, 390, 301–314.

4) Fig. 1 Please add SEM conditions in terms of resolution, voltage, etc.

Response: Thank the reviewer for your advice. The SEM conditions in terms of resolution and voltage were listed in the revised manuscript as follows: (voltage: 5.0 kV for a, 20.0 kV for b, c and d; resolution: 5000×for a, 6000×for b, c and d).

5) Line 202: The authors should also describe raw BC characteristic peaks in were. Clearly, the peak at 22.79° from BC/EVMT is attributed to BC pattern.

Response: Thank the reviewer for your comment. “A strong peak appeared to be around 2θ=22.79° in BC/EVMT, which is clearly attributed to BC pattern.” was added in the revised manuscript, line 222-223.

6) Pag. 9: The authors should include in this section some adsorption capacity values, i.e., values reported in the literature from similar materials, in order to demonstrate the novelty of these new composites.

Response: Thank the reviewer for your comment. The maximum adsorption capacity and removal efficiency of such as dye and antibiotic by similar materials were compared in table 3 in the revised manuscript.

3.3 Comparison of BC/EVMT with other related adsorbents

Vermiculite and BC based adsorbent were previously reported for removal of pollutant such as dye and antibiotic. The adsorption capacity of the similar materials was presented in Table 3. The adsorption capacity of MB and SA by different adsorbents was also compared. The as-prepared BC/EVMT displayed a competitive and satisfactory adsorption capacity and may have greater value in practical applications.

Table 3 Comparison of maximum adsorption capacity and removal efficiency of previously reported related adsorbents.

Adsorbents

Adsorbates

Maximum Adsorption Capacity (mg/g)

Removal Efficiency (%)

References

Polyamide-Vermiculite

MB

76.42

99

3

Phosphatidylcholine-Vermiculite

oxytetracycline

66.40

77

49

ciprofloxacin

93.72

96

Gellan gum/bacterial cellulose

safranin

17.57

66

50

crystal violet dye

13.49

28.5

MOF-derived carbons

SA

256

not calculated

51

chloroxylenol

588

not calculated

BC/EVMT

MB

92.16

72.7

Present work

SA

71.53

68.7

49. Liu, S.; Wu, P.; Yu, L.; Li, L.; Gong, B.; Zhu, N.; Dang, Z.; Yang, C. Preparation and characterization of organo-vermiculite based on phosphatidylcholine and adsorption of two typical antibiotics. Appl. Clay Sci. 2017, 137, 160–167.

50. Nguyen, H. T.; Ngwabebhoh, F. A.; Saha, N.; Saha, T.; Saha, P. Gellan gum/bacterial cellulose hydrogel crosslinked with citric acid as an eco-friendly green adsorbent for safranin and crystal violet dye removal. Int. J. Biol. Macromol. 2022, 222, 77–89.

51. Mondol, M. M. H. and Jhung S. H. Pore creation nanoarchitectonics from non-porous metal-organic framework to porous carbon for adsorptive elimination of sulfanilamide and chloroxylenol from aqueous solution. J. Hazard. Mater. 2022, 439, 129659.

Reviewer 3 Report

General comments

Needs major revisions especially language.

Specific comments

Introduction is short and compact but authors never justified why they have used EVMT in this work. Yes, EVMT is cheap but authors must provide proper scientific justification for choosing EVMT over other more effecient adsorbents like MOF, graphene etc.

Experimental: More details about how VMT was expanded must be provided in revised manuscript.

Results and Discussion is well written except for some minor glitches like:

a. Authors must "vertically shift" their curves in FTIR plot so that the peak shifts are clearly visible.

b. Authors used EVMT but never characterized it. XRD of unexpanded VMT must also be added in Fig. 2(b).

Author Response

Reviewer#3

General comments

Needs major revisions especially language.

Response: Thank the reviewer for your comment. We made our best efforts to carefully improve the language.

Specific comments:

1) Introduction is short and compact but authors never justified why they have used EVMT in this work. Yes, EVMT is cheap but authors must provide proper scientific justification for choosing EVMT over other more effecient adsorbents like MOF, graphene etc.

Response: Thank the reviewer for your comment.

The revised parts are as follows:

line 84-86:

It has been demonstrated that low-cost and environmental friendliness raw materials make the new composites broad application prospects.

2)Experimental: More details about how VMT was expanded must be provided in revised manuscript.

Response: Thank the reviewer for your comment. More details about how VMT expanded was provided in revised manuscript as follows:

line 118-121:

The EVMT was purchased from Lingshou, Hebei province, China. The vermiculite (industrial grade) was calcined in muffle furnace at 1000°C for 10 min, precipitated and separated, naturally dried, machine ground, and sieved at 1000 mesh to obtain pure EVMT.

3)Results and Discussion is well written except for some minor glitches like:

  1. Authors must "vertically shift" their curves in FTIR plot so that the peak shifts are clearly visible.
  2. Authors used EVMT but never characterized it. XRD of unexpanded VMT must also be added in Fig. 2(b).

Response: Thank the reviewer for your comment.

  1. FTIR plot is vertically shift to make sure that the peak shifts are clearly visible in the revised manuscript.

Fig. 2 a FT-IR spectra of BC, EVMT and BC/EVMT; b XRD spectra of BC, EVMT and BC/EVMT; c XPS spectra of BC, EVMT and BC/EVMT; d TGA curves of BC and BC/EVMT.

2. We are very sorry for our carelessness for the wrong expression. In fact, this work did not involve the use of VMT. We only use EVMT to prepare the BC/EVMT composite. The XRD pattern of EVMT has been shown in Fig. 2(b) of the original manuscript.

Reviewer 4 Report

Dear Editor, in the submitted paper an eco-friendly adsorbent based on bacterial cellulose and vermiculite composites for efficient removal of dyes and antibiotics. The paper is well organized and provides some new and interesting data. For this reason, I propose to be accepted after minor revision.

It was described in introduction that adsorption is recognized as a promising technique due to its low cost for wastewater treatment. Indeed there are several used materials for dyes removal. A recent study in this area using low cost materials has not been mentioned. Please see https://doi.org/10.3390/pr6090166

It is not clear why only 2 wt% vermiculite (EVMT) was used for the composites preparation. Was any previous study form authors and they fount that this the optimum amount?

Are any characteristics concerning the EVTM dispersity in BC?

I would expected to see also some absorption results in neat BC and not only in BC/EVMT. As the results are presented, it is not possible to see the effect of EVTM.

Author Response

Reviewer#4

Comments:

Dear Editor, in the submitted paper an eco-friendly adsorbent based on bacterial cellulose and vermiculite composites for efficient removal of dyes and antibiotics. The paper is well organized and provides some new and interesting data. For this reason, I propose to be accepted after minor revision.

1) It was described in introduction that adsorption is recognized as a promising technique due to its low cost for wastewater treatment. Indeed there are several used materials for dyes removal. A recent study in this area using low cost materials has not been mentioned. Please see https://doi.org/10.3390/pr6090166

Response: Thank the reviewer for your comment. A recent study in this area using low cost materials was cited in the revised manuscript in line 72-73 as follows: “Kyzas, etc. used very low-cost natural textile fibers as adsorbents to remove basic dye from synthetic effluents [20].”

  1. Kyzas, G.Z.; Christodoulou, E.; Bikiaris, D.N. Basic dye removal with sorption onto low-cost natural textile fibers. Processes. 2018, 6, 166.

2) It is not clear why only 2 wt% vermiculite (EVMT) was used for the composites preparation. Was any previous study form authors and they fount that this the optimum amount?

Response: Thank the reviewer for your comment. Generally, natural clay minerals contain montmorillonite, vermiculite, hectorite, zeolite, and chlorite, etc., which mainly exist in the form of hydrous aluminum phyllosilicates along with iron, magnesium, alkali metals, and other cations [1]. The previous research indicated that a 2% MMT concentration was sufficient for the saturation of the surface and matrix of BC sheets with montmorillonite particles [2]. And this article was cited in our original manuscript.

  1. Han, H.; Rafiq, M. K.; Zhou, T.; etc. A critical review of clay-based composites with enhanced adsorption performance for metal and organic pollutants, J. Hazard. Mater. 2019, 369, 780–796.
  2. Ul-Islam, M.; Khan, T.; Park, J.K. Nanoreinforced bacterial cellulose-montmorillonite composites for biomedical applications. Carbohyd. Polym. 2012, 89, 1189–1197.

3)Are any characteristics concerning the EVTM dispersity in BC?

Response: Thank the reviewer for your comment. As shown in Fig. 1b, EVMT could disperse uniformly on the BC surface.

Fig. 1 SEM images of a pure BC; b BC/EVMT; c BC/EVMT after MB adsorption; d BC/EVMT after SA adsorption (voltage: 5.0 kV for a, 20.0 kV for b, c and d; resolution: 5000×for a, 6000×for b, c and d).

4)I would expected to see also some absorption results in neat BC and not only in BC/EVMT. As the results are presented, it is not possible to see the effect of EVTM.

Response: Thank the reviewer for reminding us. The removal efficiency of BC, EVMT, and BC/EVMT for MB and SA was compared in the revised manuscript line 266-271.

As depicted in Fig. S3, the removal efficiency of BC, EVMT, and BC/EVMT for MB and SA was compared. Obviously, the BC/EVMT displayed greater adsorption capacity for both MB and SA than BC and EVMT. The EVMT alone made almost no contribution to the removal of both pollutions (<5%) because of agglomeration. The BC alone exhibits a slight removal of MB by 25.4%, and SA by 23.5%, much lower than 72.7% for MB and 68.7% for SA, which removed by the prepared BC/EVMT.

Fig. S3 Removal efficiency of BC, EVMT, and BC/EVMT for MB and SA (temperature: 30 °C, pHMB: 8, pHSA: 2, adsorbent dosage: 20 mg, [MB]0: 60 mg/L, [SA]0: 60 mg/L).

Round 2

Reviewer 1 Report

Accepted in present form, the article has been improved

Author Response

Reviewer#1

Comments:

Accepted in present form, the article has been improved

Response: Thank the reviewer for your comments.